Cross-cultural adaptation and validation of the Romanian International Knee Documentation Committee—subjective knee form

Todor Adrian 1
Vermesan Dinu 2
Haragus Horia horia.haragus@umft.ro horia.haragus@yahoo.com 2
Patrascu Jr Jenel M. 2
Timar Bogdan 3
Cosma Dan I. 1
1 Department of Orthopedics, Traumatology and Pediatric Orthopedics, University of Medicine and Pharmacy of Cluj-Napoca , Cluj-Napoca , Romania
2 Department of Orthopedics and Trauma, University of Medicine and Pharmacy of Timisoara , Timisoara , Timis , Romania
3 Department of Functional Sciences, University of Medicine and Pharmacy of Timisoara , Timisoara , Timis , Romania
Gao Liang
Electronic publication date: 2020 Feb 3
Publication date: 2020
Volume: 8
Electronic Location ID: e8448
Received 2019 Oct 10; Accepted 2019 Dec 20
Copyright: ©2020 Todor et al.
Copyright year: 2020
Copyright holder: Todor et al.
License: This is an open access article distributed under the terms of the Creative Commons Attribution License, which permits unrestricted use, distribution, reproduction and adaptation in any medium and for any purpose provided that it is properly attributed. For attribution, the original author(s), title, publication source (PeerJ) and either DOI or URL of the article must be cited.
License URL: https://creativecommons.org/licenses/by/4.0/

Keywords: Knee joint, Osteoarthritis, Meniscectomy, Arthroscopy, Anterior cruciate ligament, International knee documentation committee, Lysholm knee score, Patient reported outcome measures

Funding: The authors received no funding for this work.

==============================
Aim

We aimed to translate and cross-culturally adapt the International Knee Documentation Committee—subjective knee form (IKDC) in Romanian.

Method

The original (US) IKDC—subjective knee form was translated according to recommended guidelines. Validity was tested using Spearmans’s correlation coefficient between score sand test-retest reproducibility. Reliability and internal consistency were determined using Cronbach’s alpha coefficient and intraclass correlation coefficient (ICC).

Results

A total of 106 data sets were available for processing. The average age was 52 years and the male to female ratio was 40:66. Fifty-five subjects repeated the form after an average of 4 days. There were no floor or ceiling effects (range 3.4–74.7). There was a strong correlation between the first and repeated administration of the IKDC—subjective knee form (r = 0.816, n = 50) and moderate compared to Tegner-Lysholm knee rating scale (r = 0.506, n = 102), KOOSJR (Knee disability and Osteoarthritis Outcome Score for Joint Replacement, r =  − 0.622, n = 96), EuroqolEQ-5D-5L Index (r = 0.633, n = 100) and visual analogue scale VAS (r = 0.484, n = 99). Internal consistency was moderate with Cronbach’s alpha 0.611 (n = 102) and ICC 0.611 for average measures (95% CI 0.493–0.713).

Conclusion

The Romanian translation of the IKDC—subjective knee form is a valid, consistent and reproducible outcome measure in patients with knee pain and dysfunction.

Introduction

Knee pathology is very common. Pain and dysfunction can arise from sports injuries, trauma or degeneration and progress to chronic disability and ultimately osteoarthritis (OA). Quality evaluation of treatment outcomes also takes into account how patients perceive the results. Patient reported outcomes are therefore an integral part of clinical assessment. They provide insights on the patient’s pain relief, performance during activities of daily living, return to sports and level of competitiveness (Emery et al., 2019; Ahmad et al., 2017; Grevnerts, Terwee & Kvist, 2015).

The International Knee Documentation Committee (IKDC) was formed to establish a common ground to evaluate knee function. The IKDC originally developed an objective measurement score (clinician completed) to which a subjective knee form (patient completed) was added in 2000. It has since proved to be a commonly used form, with good psychometric properties. Because it was developed more as a knee specific rather than disease specific outcome scale, the IKDC subjective knee form is versatile, suitable to a wide range of pathologies: sports injuries, anterior cruciate ligament (ACL), meniscus, cartilage and OA  (Emery et al., 2019; Ahmad et al., 2017; Grevnerts, Terwee & Kvist, 2015).

Several IKDC subjective knee form translations are freely available on the American orthopedic society for sports medicine (AOSSM) webpage  (AOSSM, 2019). It has been translated and culturally adapted in several languages but not Romanian (AOSSM, 2019; Çelik et al., 2014; Koumantakis et al., 2016; Huang et al., 2017).

We therefore aimed to perform the translation, cross-cultural adaptation and validation of the International Knee Documentation Committee—subjective knee form (IKDC) in patients with knee pain and dysfunction.

Materials and Methods

The original 10 question IKDC—subjective knee form was retrieved from the developer’s website (AOSSM, 2019). Response options vary among items: questions 6 dichotomizes response into yes/no; questions 1, 4–8, and 9 use 5-point Likert scales and questions 2, 3 and 10 use 11-point numerical rating scales. The English (US) IKDC form was translated and culturally validated using the ISPOR (International Society for Pharmacoeconomics and Outcomes Research) principles of good practice for the translation and cultural adaptation process. The process of translation was straight forward (Wild et al., 2005; Haragus et al., 2018). Two separate translators, proficient in both English and Romanian performed the forward translations with discretionary adnotations. These two forms were reviewed by the authors and we decided on a common form. This later was backward translated from Romanian to English by a native English speaker, proficient in Romanian and the result compared and contrasted to the original English (US) IKDC form. Two authors interviewed 5 subjects for the process of cognitive debriefing. Two issues were raised, one of questionairre construct and one semantic. The IKDC subjective knee form has several items (1, 5, 7, 8, 9-partial) aimed at diferentiating the ability to perform intense physical activity, such as one is expected to encounter while playing sports. For old and sedentary patients, these items cannot diferentiate well between the normal state of activity and a handicap due to injury or disease. Item 10 was found to be the most complex regarding translation by the authors. It was also found to easily be unclear or confusing to answer by the pretest subjects and possibly requires more attention in order to be accurately answered. Finally the Romanian translated form was proofread by a linguistic expert.

We screened adult patients with chronic knee pain and/or dysfunction, evaluated in our University affiliated Emergency clinical county hospital from MarchtoDecember 2018. Diagnosis was supported by patient history, clinical examination, X-rays, MRI (magnetic resonance imaging-where available) and arthroscopic exploration (if applicable). Indication for arthroscopy/ meniscectomy was made by the treating physician (orthopedic surgeon) based on current guidelines using a standard technique (Beaufils et al., 2017; Todor, Caterev & Nistor, 2016). Chronicity was defined as first onset of symptoms at least 4 weeks prior examination in order to maintain homogeneity. Cases with acute trauma, fractures, advanced OA requiring arthroplasty, ACL reconstruction, patellar instability, tumors and septic arthritis were excluded. The study was conducted in accordance with the Declaration of Helsinki and the protocol was approved by the Emergency clinical county hospital ‘Pius Brinzeu’ Timisoara ‘Local ethics committee for scientific research’. All patients gave their informed consent for inclusion before they participated in the study.

The subjects completed Romanian translations of IKDC—subjective knee form and KOOSJR (Knee disability and Osteoarthritis Outcome Score for Joint Replacement) and Euroqol EQ-5D-5L Index (converted using the UK tariff) and visual analogue scale (VAS) when seen during clinics  (Lyman et al., 2016; EuroQol, 2019). The examining physician (orthopedic surgeon/physical therapist) then completed the Tegner-Lysholm knee rating scale  (Orthopaedic Scores, 2019).

Construct validity was tested using Spearmans’s correlation coefficient between the tested scores. Internal consistency was determined using Cronbach’s alpha coefficient and test retest reliability using the intra class correlation coefficient (ICC, two-way mixed effects model)  (Çelik et al., 2014; Koumantakis et al., 2016; Huang et al., 2017; Haragus et al., 2018). For all tests, higher values were associated with better results. Data was analyzed using SPSS v17 statistical software package (SPSS Inc, Chicago, IL, USA).

Results

One hundred and six (106) data sets were proccesed, out of 110 completed. 19 patients declined participation. 97 underwent knee arthroscopy. Average age was 52 (range 21–83) years and male to female ratio 40:66 (1:1.67). 55 subjects repeated the IKDC—subjective knee form after an average of 4 days (range 1–7). There were no floor or ceiling effects for both IKDC—subjective knee form scores (min 0–max 100), which ranged from 3.4 to 74.7 for the first and 4.6–74.7 for the second.

Twelve consecutive patients were interviewed and timed at the first completion of the IKDC—subjective knee form. Two required glasses to read the questionnaire. The patients completed the score in an average of 3 min and 4 s and found it clear and straight forward. Nine estimated that they could complete the questionnaire through mail and phone and even email or tablet with assistance from family members.

There was a strong correlation between the first and repeated administration of the IKDC—subjective knee form (r = 0.816, n = 50) and moderate compared to Tegner-Lysholm knee rating scale (r = 0.506, n = 102), KOOSJR (r =  − 0.622, n = 96), EQ-5D-5L Index (r = 0.633, n = 100) and VAS (r = 0.484, n = 99) (see Table 1, Figs. 1 and 2).

Table 1 Spearman’s rho correlation coefficients between the tested scores.

Correlations between the two IKDC (International Knee Documentation Committee—subjective knee form) scores, Tegner-Lysholm knee rating scale, KOOSJR (Knee disability and Osteoarthritis Outcome Score for Joint Replacement) and EQ-5D-5L Index and VAS (Visual analog scale), presented as coefficient/p value and number of subjects.

	IKDC	ikdc2	Lysholm	KOOSJR	EQ-5D-5L	VAS	
IKDC	1.000	.816**	.506**	−.622**	.633**	.484**	
.	.000	.000	.000	.000	.000	
103	50	102	96	100	99	
ikdc2	.816**	1.000	.392**	−.670**	.586**	.568**	
.000	.	.004	.000	.000	.000	
50	54	53	48	50	49	
Lysholm	.506**	.392**	1.000	−.546**	.513**	.436**	
.000	.004	.	.000	.000	.000	
102	53	105	95	99	98	
KOOSJR	−.622**	−.670**	−.546**	1.000	−.562**	−.567**	
.000	.000	.000	.	.000	.000	
96	48	95	96	96	92	
EQ-5D-5L	.633**	.586**	.513**	−.562**	1.000	.591**	
.000	.000	.000	.000	.	.000	
100	50	99	96	100	96	
VAS	.484**	.568**	.436**	−.567**	.591**	1.000	
.000	.000	.000	.000	.000	.	
99	49	98	92	96	99	
Notes.

** Correlation is significant at the 0.01 level (2-tailed).

Figure 1 Correlation between IKDC and Tegner-Lysholm.

Moderate correlation between IKDC (International Knee Documentation Committee—subjective knee form) and Tegner-Lysholm knee rating scale.

Figure 2 Correlation between IKDC and EQ-5D Index.

Moderate correlation between IKDC (International Knee Documentation Committee—subjective knee form) and EQ-5D-5L Index.

Internal consistency and test-retest reliability were moderate. For the first IKDC—subjective knee form, Cronbach’s alpha was 0.611 (n = 102) and ICC 0.611 for average measures (95% CI [0.493–0.713]). For the retest, Cronbach’s alpha was 0.593 (n = 55) and ICC 0.593 for average measures (95% CI [0.418–0.734]).

Discussion

The Romanian translated and culturally adaptated IKDC—subjective knee formproved valid, reliable, consistent and reproducible in patients with non-acute knee pain and dysfunction. However, internal consistency and test-retest reliability were moderate, compared to recently published literature regarding IKDC—subjective knee form translations: TurkishCronbach’s 0.89 and ICC 0.91; Greek Cronbach’s 0.87 and ICC 0.95; Chinese Cronbach’s 0.87 and ICC 0.97 (Çelik et al., 2014; Koumantakis et al., 2016; Huang et al., 2017).

Most of our valid entries (91.5%) were patients who underwent knee arthroscopy. Out of them, by far the main indication was meniscectomy. Treatment of symptomatic knees with meniscus tears can be controversial, however it is still one of the most routinely performed orthopedic surgical procedure (Grevnerts, Terwee & Kvist, 2015; Beaufils et al., 2017; Todor, Caterev & Nistor, 2016). Cartilage surgery is still under development, with cautiously optimistic predictions for the future (Fodor et al., 2018). The transition from cartilage defects and meniscus tears to early OA is many times difficult. For incipient knee degeneration, definition, symptoms, magnetic resonance imaging and outcome measures are not yet standardized. Nevertheless, for this subpopulation the IKDC—subjective knee form, the KOOSJR, Tegner-Lysholm knee rating scale and Euroqol EQ-5D-5L are among the most commonly used and supported patient reported outcomes  (Emery et al., 2019; Grevnerts, Terwee & Kvist, 2015; Jones et al., 2016).

The ACL is arguably the most commonly reconstructed ligament in the human body. It is the main stabilizer against anterior tibial translation, with functional importance in sports. The aforementioned outcome measures apply also in cruciate ligament reconstruction  (Ahmad et al., 2017; Grevnerts, Terwee & Kvist, 2015; Todor, Nistor & Caterev, 2019). We deliberately excluded ACL surgeries to maintain group homogeneity and target the non-acute pain, meniscus and early degeneration subpopulation.

Our study has several limitations. We did not use the entire IKDC questionnaire, nor the complete KOOS (Knee disability and Osteoarthritis Outcome Score). Instead, we opted for the Tegner-Lysholm knee rating scale, a versatile and simple outcome measure, widely used in sports and arthroscopy, that can be completed both by the clinician as well as the patient (Ahmad et al., 2017; Grevnerts, Terwee & Kvist, 2015). The shorter version of the KOOS developed for joint replacement was also favored as a shorter, simpler form for early knee OA. We based our decision on the considerable overlap of patients with early radiographic OA with those with advanced degeneration (Emery et al., 2019; Lyman et al., 2016; Jones et al., 2016). For the general wellbeing, the Euroqol EQ-5D-5L is a free, widely used patient questionnaire (Emery et al., 2019; Sørensen et al., 2019). However, to quantify impairment we had too use the UK tariff as the best approximation since there is no conversion available for Romania. In the original development study of the IKDC, the authors used the Short Form 36 (SF-36) subscales as comparators, which became the standard for subsequent validations (Grevnerts, Terwee & Kvist, 2015; AOSSM, 2019; Çelik et al., 2014; Koumantakis et al., 2016; Huang et al., 2017). Compared to the 5 item EQ-5D-5L, the SF-36, commercially administered by the RAND Corporation (Santa Monica, CA, USA) has 36, divided into 8 sections. The issues listed above probably had 2 effects: they significantly reduced the effort required to fill out the forms at the cost of decreasing the validity correlation strength. We felt this to be a fair trade as well as maintain a current trend towards the reduction of item number deemed relevant in regularly used patient reported outcomes. It is likely that simplified versions of highly referenced scores and computer adapted technology will usher in the implementation of patient reported outcomes from the research field to routine clinical practice  (Lyman et al., 2016; Sørensen et al., 2019; Onofrei et al., 2019).

Conclusions

The Romanian translation of the IKDC—subjective knee form is a valid, consistent and reliable outcome measure in patients with knee pain and dysfunction. However, internal consistency and test-retest reliability were moderate compared to published literature.

Supplemental Information

Supplemental Information 1 Romanian IKDC—subjective knee form.

Click here for additional data file.

Data S1 Raw data

Click here for additional data file.

Bogdan Deleanu, MD, PhD; IoanaGeorgescu, MD; Andrei Ghiorghitoiu, MD; Musat Roxana, Med Stud.

Additional Information and Declarations

Competing Interests

Author Contributions

Human Ethics

Data Availability

The authors declare there are no competing interests.

Adrian Todor, Horia Haragus and Dan I. Cosma conceived and designed the experiments, authored or reviewed drafts of the paper, and approved the final draft.

Dinu Vermesan and Jenel M. Patrascu Jr performed the experiments, authored or reviewed drafts of the paper, and approved the final draft.

Bogdan Timar analyzed the data, prepared figures and/or tables, and approved the final draft.

The following information was supplied relating to ethical approvals (i.e., approving body and any reference numbers):

Emergency clinical county hospital ‘Pius Brinzeu’ Timisoara ‘Local ethics committee for scientific research’ 177/ 2019.

The following information was supplied regarding data availability:

The raw measurements are available in the Supplemental Files.

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
