# Peer review of "Cross-cultural adaptation and validation of the Romanian International Knee Documentation Committee—subjective knee form"

_PeerJ, doi:10.7717/peerj.8448_

## Round 0.1 · original submission · Minor Revisions

Dear Authors,

Your manuscript, “Cross-cultural adaptation and validation of the Romanian International Knee Documentation Committee – subjective knee form”, has been reviewed by our ad-hoc reviewer with expertise in the field who feel that your manuscript in the current status can not be published in the PeerJ before a Minor Revision. Their commentary is attached below for your information.

Thank you for the opportunity to review your manuscript at this time.

Sincerely,

Liang Gao, MD, PhD
Academic Editor, the PeerJ
https://peerj.com/LiangGao/

·

Basic reporting

The article structure, figures and tables are all of professional level, however some specific comments for figures and tables that are unnecessary or incorrectly reported are provided below.
Some references should also be added.
The cross-cultural adaptation procedure should be briefly described.

Experimental design

Methods not described with sufficient detail & information to replicate in some respect, and this is pointed out to the authors.

Validity of the findings

All underlying data have been provided; they are robust, however, some departures from statistical soundness are pointed out.

Additional comments

1. The authors state they had ‘106 valid sets available for processing’ in all statistical analysis presented n was lower. Apparently there were not 106 complete sets, as attested in the appended Excel file.


2. The ISPOR criteria for cross-cultural adaptation should be referenced. A reference you could use is the following:
Wild D, Grove A, Martin M, Eremenco S, McElroy S, Verjee-Lorenz A, Erikson P; ISPOR Task Force for Translation and Cultural Adaptation. Principles of good practice for the translation and cultural adaptation process for patient-reported outcomes (PRO) measures: report of the ISPOR Task Force for translation and cultural adaptation. Value Health. 2005;8(2):94-104.
However, a brief description of the cross-cultural adaptation procedure you followed should also be presented.

3. Lines 100-1: Why was chronicity defined in case symptoms persisted over 4 weeks? Please provide reference.

4. Line 112: Why was the Spearman’s correlation used instead of Pearson’s? Please provide statistical reasoning and reference(s).

5. Line 113-4: The statistics for internal consistency and reliability are not clearly stated. Please change. Also, reference 14 is not a primary reference for ICC selection.

6. Lines 115-6: Test-retest reproducibility should be tested with ICC, not Spearman’s!

7. Line 143 & 144: ICC is not a statistic for Internal Consistency. Please correct.

8. Table 1 – page 14: In the table the word Index probably refers to the EuroQuol 5D-5L. Please change. Also, the use of Spearman’s Rho for test-retest reliability is not the appropriate statistic-ICC is.

9. Figure 1 – page 15: 2 different legends are provided for this figure. Again, the use of Spearman’s Rho for test-retest reliability is not the appropriate statistic-ICC is, as it takes into account a systematic difference that may be present between-ratings. I suggest this figure to be omitted altogether.

10. Figure 2 &3 – pages 16 & 17: Again, 2 different legends are provided for these figures.

Reviewer 2 ·

Basic reporting

Good English language, with relevant literature references.

Experimental design

The design of the study is respecting all the required steps for validating a functional score.
The statistic method was appropriately performed.

Validity of the findings

It is important to have clinical scores validated in any language, by using those scores we can objectively assess and compare our results.
I consider that this article is respecting all the requirements for a validation of a clinical score.

Additional comments

The Romanian translation and validation of the IKDC score is a useful assessment tool for all the practitioners in this country .
The study is respecting all the requirements of a validation process.

---

## Round 0.2 · accepted · Accept

Congratulations! Your manuscript - Cross-cultural adaptation and validation of the Romanian International Knee Documentation Committee – subjective knee form - has been Accepted for publication.

·

Basic reporting

Reporting is clear and concise. References are now correct.

Experimental design

Up to standard. The methodology for cross-cultural adaptation of patient-reported outcomes is clearly portrayed.

Validity of the findings

All data have been provided, correctly analysed and clearly presented. In table 2 the word 'Index' on the vertical axis can be replaced with Euroqol5d5l for clarity and parity.

Additional comments

Thank you for addressing all comments.